# Exploratory Factor Analysis (EFA) of the Short Functional Geriatric Evaluation (SFGE) to Assess the Multidimensionality of Frailty in Community-Dwelling Older Adults

**DOI:** 10.3390/ijerph20054129

**Published:** 2023-02-25

**Authors:** Giuseppe Liotta, Grazia Lorusso, Olga Madaro, Valeria Formosa, Leonardo Emberti Gialloreti, Clara Donnoli, Fabio Riccardi, Stefano Orlando, Paola Scarcella, Joao Apostolo, Rosa Silva, Carina Dantas, Willeke van Staalduinen, Vincenzo De Luca, Maddalena Illario, Susanna Gentili, Leonardo Palombi

**Affiliations:** 1Department of Biomedicine and Prevention, University of Rome Tor Vergata, 00133 Rome, Italy; 2“Long Live the Elderly!” Program, Community of Sant’Egidio, 00153 Rome, Italy; 3The Health Sciences Research Unit: Nursing (UICISA: E), Nursing School of Coimbra, 3000-076 Coimbra, Portugal; 4SHINE 2Europe, 3030-163 Coimbra, Portugal; 5AFEdemy—Academy on Age-Friendly Environments in Europe, 2806 ED Gouda, The Netherlands; 6Department of Public Health, University of Naples “Federico II”, 80131 Naples, Italy; 7Aging Research Center, Department of Neurobiology, Care Sciences and Society, Karolinska Institutet and Stockholm University, 17177 Stockholm, Sweden

**Keywords:** EFA, SFGE, frailty, older adults, multidimensional evaluation, biopsychosocial frailty, assessment tool

## Abstract

The Short Functional Geriatric Evaluation (SFGE) is a multidimensional and short questionnaire to assess biopsychosocial frailty in older adults. This paper aims to clarify the latent factors of SFGE. Data were collected from January 2016 to December 2020 from 8800 community-dwelling older adults participating in the “Long Live the Elderly!” program. Social operators administered the questionnaire through phone calls. Exploratory factor analysis (EFA) was carried out to identify the quality of the structure of the SFGE. Principal component analysis was also performed. According to the SFGE score, 37.7% of our sample comprised robust, 24.0% prefrail, 29.3% frail, and 9.0% very frail individuals. Using the EFA, we identified three main factors: psychophysical frailty, the need for social and economic support, and the lack of social relationships. The Kaiser–Meyer–Olkin measure of sampling adequacy was 0.792, and Bartlett’s test of sphericity had a statistically significant result (*p*-value < 0.001). The three constructs that emerged explain the multidimensionality of biopsychosocial frailty. The SFGE score, 40% of which is social questions, underlines the crucial relevance of the social domain in determining the risk of adverse health outcomes in community-dwelling older adults.

## 1. Introduction

The median age in the EU27 is projected to increase by 4.5 years between 2019 and 2050, reaching 48.2 years [1]. In Italy, more than 7 million people (7,058,755) are over 75 years old, 11.7% of the total population [2], positioning itself at the top of the EU pyramid [3]. The growth of the ageing population has brought the issue of biopsychosocial frailty to the attention of public health [4]. Biopsychosocial frailty, a synthetic indicator of the need for care, could be used to plan out-of-hospital care in order to maximise the efficacy and effectiveness of community services. It is associated with the more intense use of both hospital and out-of-hospital services, thus constituting a strong quantitative and qualitative predictor of these cases [5]. Furthermore, frailty is correlated with adverse outcomes, such as falls, disability, hospitalisation, institutionalisation, and death [6]. Consequently, as the frailty level increases, so do hospital and nonhospital services. This affects the cost of health and social services, increasing public spending and causing frailty to become a major public health problem [7]. To foster and promote the social participation and independent living of older adults, a better alignment among health, social care, built environments (e.g., houses, cities, transportation systems), and resources such as information and communication technologies (ICT) is needed [8].

Frailty was defined by Gobbens et al. as a dynamic state affecting an individual who experiences losses in one or more domains of human functioning (physical, psychological, social) that are caused by the influence of a range of variables, and which increases the risk of adverse outcomes [9]. This definition is the result of a large global consensus conference involving experts from several countries investigating frailty through the physical, psychological, and socioeconomic domains [10]. The lack of integration in care processes, not allowing for communication between different professions and communities in care and health settings, causes inadequacies in clinical and public health control regarding frailty management. Widescale frailty screening programs would allow for the early detection of the need for supporting functional capacities, and the adaption of environmental factors and health promotion strategies [11]. From a public health perspective, to render the assessment of frailty functional on a wide scale (millions of people assessed at least once a year), short and predictive tools should be validated [12]. However, available short tools lack a strong social domain, which is crucial when we are dealing with community-dwelling older adults to address care intervention [12]. The Short Functional Geriatric Evaluation (SFGE) reflects these characteristics; it is the short version of the Functional Geriatric Evaluation (FGE) [13].

The FGE is a scale validated in Italy by Palombi et al. [13] and derived from the Geriatric Functional Rating Scale [14]. SFGE is a multidimensional, quick, and easy-to-use questionnaire to assess frailty in community-dwelling older people [7]. In this scale, the socioeconomic aspects are represented very well, composing 60% of the items and 40% of the final score, while the remaining items are related to psychophysical domains. This questionnaire had acceptable short-term sensitivity (90.4%) and a medium–low specificity (78.3%) for the assessment of frailty, similarly to other available tools devoted to quickly assessing frailty in community-dwelling older adults [15]. The final SFGE score stratifies elderlies according to four levels of frailty: robust, prefrail, frail, and very frail. The SFGE is predictive for adverse events, such as hospitalisation, institutionalisation, and death. SFGE is significantly associated with a risk of death that increases progressively from prefrail to very frail individuals if compared to robust ones [16]. The SFGE is also highly predictive of hospitalisation and institutionalisation, as the risk of these increases with SFGE level. Furthermore, all these results were confirmed in individuals who did not have psychophysical impairments, but showed socioeconomic frailty. A lower socioeconomic status in older adults is associated with frailty and adverse health outcomes [17]. This study conducts exploratory factor analysis (EFA) to assess the latent factors of SFGE. This paper will be followed by another work to clarify the construct validity of the SFGE.

## 2. Materials and Methods

This is a cross-sectional study carried out using secondary analysis data collected from January 2016 to December 2020 in nine Italian cities: Brindisi, Catania, Civitavecchia, Ferentino, Genoa, Naples, Novara, Rome, and Sassari. Our sample was composed of 8800 community-dwelling older people over 65 participating in the “Long Live the Elderly!” (LLE) program. Data were analysed from February to December 2021. The LLE programme [18] was set up by the community of Sant’Egidio in 2004 to reduce the negative health consequences caused by heatwaves in older people living in Rome. Later, the programme was extended to other cities. Previously trained nonhealth professionals contacted the older people through phone calls to assess biopsychosocial frailty and to counteract social isolation, which is a prominent risk factor for death and hospitalisation during heatwaves, by implementing person-centred interventions that lasted all year round. During the phone calls, informed consent was obtained from all subjects involved in the study. The operators then administered the SFGE questionnaire [19] to stratify the population according to the level of frailty. Frail people were candidates for an individualised care plan based on their needs, ascertained through a supplementary interview/discussion involving the client, the caregiver(s), the social worker responsible for that client within the LLE programme, and other professionals according to the client’s needs. The study was approved by the independent ethics committee of the University of Tor Vergata, Rome (R.S. 60/17) and was conducted following the principles of the Declaration of Helsinki.

### 2.1. Measurements

The SFGE is a questionnaire composed of 13 items that assesses multidimensional frailty in community-dwelling older people, and it comprises the psychophysical and socioeconomic domains [7]. Nonhealth professionals with a high-school diploma can administer it in about 10 min. The total score is used to stratify older people into four levels: robust (score ≤ 0), prefrail (1–2), frail (3–9), and very frail (≥10).

### 2.2. Statistical Analysis

All the statistical analyses were conducted considering the score of each item of the SFGE. The items were reduced from 13 to 12 because Questions 7 and 8 were merged (Appendix A). The questions were both about economic difficulties, but the critical item was Question 7. The item was divided into three response grades that corresponded to making it to the end of the month on one’s economic resources (score 0), having problems paying for a person to help at home when needed (score 1), and having problems buying necessary food (score 2). In this sense, combining the two questions into a single answer expresses the logic of the questionnaire, which wants to know the presence and extent of economic difficulties. For practical reasons, the results of the individual items were all converted into positive values and then set to an increasing value corresponding to a whole number from 1 onwards. EFA was carried out to identify the goodness of the structure of the SFGE through the latent factors that could describe frailty and its multidimensional characteristics. Principal component analysis was performed with varimax rotation and Kaiser normalisation. We decided to perform principal component analysis because the sample consisted of more than 15 people, the missing data were less than 1%, and we used pairwise deletion. Kaiser’s criteria (eigenvalues greater than 1) were used as factor retention methods [20]. Statistical analyses were performed using IBM SPSS Statistics version 26 for Windows (SPSS Inc., Chicago, IL, USA). *p*-values < 0.05 were considered statistically significant.

## 3. Results

Our sample was composed of 8800 people over 65—67.2% males and 32.8% females. The mean age was 84.3 (SD ± 4.7) for males and 84.5 (SD ± 5.5) for females. Most people came from Rome (n = 4331, 49.1%), Naples (n = 1874, 21.3%), and Novara (n = 1218, 13.8%) (Table 1).

According to the SFGE score, 37.7% of our sample were robust (n = 3319), 24.0% prefrail (n = 2108), 29.3% frail (n = 2577), and 9.0% very frail (n = 796) individuals.

Table 2 shows how people answered the 12 questions of the SFGE. The higher the question score was, the greater the tendency to frailty.

In the correlation matrix, there were no significant correlation coefficients that suggested the presence of multicollinearity (Appendix A). Through principal component analysis with the varimax rotation method, we extracted 3 factors with an eigenvalue > 1.0 according to the Kaiser–Guttman rule. The three identified latent factors were psychophysical frailty (Factor 1), the need for social and economic support (Factor 2), and the lack of social relationships (Factor 3) with eigenvalues of 2.94 (24.53%), 1.39 (11.58%) and 1.10 (9.21%), respectively. The Kaiser–Meyer–Olkin measure of sampling adequacy was 0.792, a value that supports the use of EFA and minimises the risk of biases due to sample size issues. Bartlett’s test of sphericity showed a statistically significant result (*p*-value < 0.001). The initial factor loadings before rotation were all above 0.3, which means that all loading factors contributed to assessing the underlying dimensions. In the rotated factor matrix, we did not consider factor loadings lower than |0.30| (Table 3 and Table 4).

In conclusion, we found via EFA that six items (Q6, Q8–Q12) had their highest loading from Factor 1 (psychophysical frailty), three items (Q3, Q4, Q7) from Factor 2 (need of social and economic support), and three items (Q1, Q2, Q5) from Factor 3 (lack of social relationship). Although Items Q1 and Q3 were both loaded onto two factors (Factors 2 and 3), we associated them with Factors 3 and 2, respectively, because of their greater affinity to the domain they represent. In Q1 and Q3, we also considered the values of the items that positively loaded onto the selected factor.

## 4. Discussion

We studied the construct validity of SFGE, which can be considered one of the most important psychometric properties of questionnaires, through EFA. It assesses how well a test measures and represents the construct for which it was conceived [21].

Using the EFA conducted on the SFGE, we identified three main factors: psychophysical frailty (Factor 1), the need for social and economic support (Factor 2) and the lack of social relationships (Factor 3). Although EFA showed that Items Q1 and Q3 were loaded onto two factors (Q1: Factor 2 and 3; Q3: Factors 1 and 2), we decided not to remove them, and to associate Q1 with Factor 3, which relates to social support, and Q3 with Factor 2, which describes the socioeconomic dimension. The reason for this association is inherent in the items themselves: in fact, a higher age (Q1) is reasonably associated with more severe physical impairment, which can result in limitations to participate in events or interactions outside the home environment, which is associated with poor social relationships; hence, this has greater affinity with Factor 3, while loneliness (living alone, Q3) is more associated with the need for socioeconomic support; therefore, it is associated with Factor 2. Factor 1 was associated with items investigating psychophysical frailty related to the necessity of receiving home care from health services or participating in daily centres (Q6), worse psychological status (motivation and energy—Q8), not being able to bathe or shower without the help of others (Q9), not being able to leave one’s home (Q10), and being bedridden (Q11) or severely confused (Q12). Of all these items, only Q6 was inversely correlated with psychophysical frailty, indicating that persons at home with worse psychophysical status are more likely to receive home care services. Factor 2 was related to items concerning the need for socioeconomic support. In fact, these questions analyse living arrangements as living alone, with the spouse, with assistance, or with others (Q3), having someone available to help in case of need (Q4), and the economic situation as receiving a pension or a subsidy and having enough money for monthly expenses (Q7). The lack of socioeconomic resources plays an important role in determining and worsening frailty in older people, and it increases the risk of hospitalisation [19], institutionalisation [22], and death [23]. Factor 3 was related to items regarding the lack of social relationships composed of older age (Q1), lower educational level (Q2), and a lower level of involvement in social activities or groups (Q5). Social isolation involves the loss of a social network that is essential to not feeling lonely or abandoned, and for maintaining an active lifestyle that prevents possible negative outcomes related to frailty [24]. Frailty in people living in the community is strongly related to the interaction between psychophysical status and social domain. In fact, we are not simply dealing with a clinical syndrome that evidently exists, but mainly with the risk of negative outcomes that are related to domains that are intertwined. The three constructs emerging from the EFA explain the multidimensionality of frailty, and the final SFGE score is strongly related to the incidence of negative outcomes, namely, death, hospitalisation, and institutionalisation [16]. The SFGE score comprises 40% questions about the social and economic status of the interviewees, which is a high rate compared to other questionnaires investigating frailty in older people. Many other questionnaires assess multidimensional frailty considering socioeconomic features; for example, the Tilburg Frailty Indicator (TFI) [25] is a validated and frequently used scale in which the socioeconomic part is about 20% of the total items. Other interesting scales can be considered the SUNFRAIL tool [26], of which almost 22% are socioeconomic questions, and AGILE [27], which is composed of socioeconomic questions by almost 20%. Recently, the Health Assessment Questionnaire for Older Adults in Japan was tested with both EFA and confirmatory factor analysis, again demonstrating the robustness of the multidimensional approach to frailty, and the relevance of combining the two statistical techniques to achieve reliable results in this field. However, again, the contribution of social domains to the questionnaire score was less than 20% [28]. 

Therefore, in public healthcare, it is difficult to choose a better scale to use because of the many characteristics of existing questionnaires, such as psychometric properties, the features of the sample considered in previous studies, and the predictive validity of adverse outcomes. In this regard, the socioeconomic aspects assessed with the SFGE play a fundamental role in establishing the frailty level of community-dwelling older adults’. In addition, a recent systematic review indicated that economic difficulties play a crucial role in determining frailty in older people. This study shows that the financial problems of older adults also reduce their social network and consequently their quality of life [29]. The link between education and economic status is also well-known. Lastly, living arrangements are intuitively a constituent of the social dimension, such that they contribute to greater or lesser frailty in this regard.

Using SFGE, frailty can be assessed on a large scale. This questionnaire can be used to screen older populations to plan public health interventions, prevent adverse outcomes, and reduce hospitalisation rates.

As a limitation, the questionnaire can be expected to provide only an initial assessment of frailty from which indications for preventive or curative action could be derived if necessary. Moreover, the method used in this paper is under debate due to some internal limitations that should be taken into consideration when results are discussed. The use of EFA with varimax rotation is still debated: from a careful review of the literature [28,30,31,32,33], it seems acceptable even if the variables are not all purely quantitative: in fact, some items have yes/no responses that we considered the ordinal variable in the framework that is the final summed-up score. Confirmatory factor analysis should be the next step to provide deeper insights about the effectiveness of the SFGE to assess frailty at the community level.

## 5. Conclusions

The model for the SFGE based on EFA was suitable, indicating that the SFGE had construct validity and could be used to assess biopsychosocial frailty in community-dwelling older adults.

In light of this EFA, SFGE deeply assesses the social domain, as the associated items were found in Factors 2 and 3.

Furthermore, tools that assess the social domain of frailty allow for promoting a proactive living style and creating personalised care plans that are not exclusively health-related, but with proper social and health integration.

The application of the SFGE could be implemented using an ICT approach [34,35] to plan public health interventions, create individual care plans, and reduce the costs of health and social services.

Despite some limitations, the SFGE has potential to be used as a measurement tool to assess frailty in future studies and daily practice. To exploit its potentialities, confirmatory factor analysis is needed.

## Figures and Tables

**Table 1 ijerph-20-04129-t001:** City of provenance of the sample.

City	Number of Patients	Rates (%)
Rome	4331	49.2
Naples	1874	21.3
Novara	1218	13.8
Genoa	671	7.6
Sassari	299	3.4
Brindisi	155	1.8
Catania	151	1.7
Civitavecchia	53	0.6
Ferentino	48	0.5
Total	8800	100.0

**Table 2 ijerph-20-04129-t002:** Distribution of answers to the questionnaire, n (%).

Item	Response	Number of Respondents (no. (%))
Q1: age	<75 75–85 >85	211 (2.4%) 5069 (57.6%) 3520 (40.0%)
Q2: education	None/primary Secondary/degree	6343 (72.1%) 2457 (27.9%)
Q3: cohabitants	Alone With spouse Spouse age <75 Spouse age 75–85 Spouse age >85 Missing With a paid assistant and others	2587 (29.4%) 4096 (46.5%) 309 (7.5%) 2314 (56.5%) 722 (17.6%) 751 (18.3%) 2117 (24.0%)
Q4: Informal/formal social network: in case of need, is there someone you can count on?	Yes, for as long as necessary Yes, occasionally No	6605 (75.0%) 1789 (20.3%) 406 (4.6%)
Q5: Informal/formal social network: are you involved in social activities or group?	Yes No	1638 (18.6%) 7162 (81.4%)
Q6: Informal/formal social network: are you receiving formal care services?	Yes No	972 (11.0%) 7828 (89.0%)
Q7: Economic situation: is your monthly income enough to get to the end of the month?	Yes With difficulty No If you answered no or with difficulty, what have you had problems with in the last month? To buy food, medicine, or clothes, to pay bills, or to pay a person to help me with activities of daily living To pay for household help	6999 (79.5%) 1705 (19.4%) 96 (1.1%) 1136 (63.1%) 665 (36.9%)
Q8: Psychological condition: energy and motivation	Normal Hypoactive/hyperactive	7777 (88.4%) 1023 (11.6%)
Q9: Health/functional status: able to use the shower or bath independently	Yes No	6543 (74.3%) 2257 (25.7%)
Q10: Health/functional status: leaves the house	Yes No	6983 (79.3%) 1817 (20,7%)
Q11: Health/functional status: bedridden	Yes No	466 (5.3%) 8334 (94.7%)
Q12: Health/functional status: confused	Yes No	799 (9.1%) 8001 (90.9%)

**Table 3 ijerph-20-04129-t003:** Results of exploratory factor analysis for the Short Functional Geriatric Evaluation (factor matrix).

Factors			
Items	1	2	3
Q1	0.358	−0.084	0.426
Q2	0.199	0.532	0.346
Q3	−0.440	0.329	−0.173
Q4	−0.177	0.532	−0.417
Q5	0.104	0.501	0.516
Q6	−0.445	0.168	0.378
Q7	0.099	0.634	−0.264
Q8	0.610	0.122	−0.244
Q9	0.788	0.000	0.069
Q10	0.745	0.072	0.084
Q11	0.640	−0.062	−0.148
Q12	0.613	0.057	−0.171

**Table 4 ijerph-20-04129-t004:** Results of exploratory factor analysis for the Short Functional Geriatric Evaluation (rotated factor matrix).

	Factors		
Items	1	2	3
Q1	0.208	−0.417	0.316
Q2	0.098	0.122	0.646
Q3	−0.356	0.452	0.012
Q4	−0.025	0.698	0.033
Q5	−0.045	0.012	0.726
Q6	−0.534	−0.017	0.288
Q7	0.193	0.611	0.266
Q8	0.658	0.106	0.038
Q9	0.728	−0.225	0.212
Q10	0.685	−0.172	0.262
Q11	0.652	−0.097	−0.014
Q12	0.637	0.011	0.045

## Data Availability

The data presented in this study are available on request from the corresponding author. The SFGE questionnaires were distributed among the centres. The electronic data were transmitted with modern cryptography systems over the web, and stored in a locked, password-protected computer.

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
