# Peer review of "Exploratory Factor Analysis (EFA) of the Short Functional Geriatric Evaluation (SFGE) to Assess the Multidimensionality of Frailty in Community-Dwelling Older Adults"

_ijerph, 2023, doi:10.3390/ijerph20054129_

Round 1
Reviewer 1 Report
The study is important and provides evidence about the psychometric property of a tool. There is need to make revisions to each section to meet the criterion for scientific writing and publication.
Introduction:
It is well-written and explaining the rationale for such tools. There are few points which should be addressed to make introduction more robust.
Comment 1. The first paragraph is too long and it covers three ideas which can be segregated. From line 48, should start next paragraph which is focusing on the definition of the construct of frailty and rather starting the sentence with “According to the definition …..” can begin this paragraph with “Frailty has been defined as ……”
Comment 2. Line 52-53 “Precisely for its multidimensional characteristic, frailty should be investigated through the physical, psychological, and socio-economic domains.” Before stating this should provide the base has it not been measured previously as a multidimensional construct?
Comment 3. Line 54-55 “The lack of integration in care processes, not including different professions and communities in different care and healing settings, causes a no link in frailty management between clinical and public health management”. This need rephrasing as it’s not clear in terms of comprehension of the point made by authors’. This point should also be elaborated further.
Comment 4: Line 60 “The Short Functional Geriatric Evaluation (SFGE) reflects these characteristics, in fact, it is the short version of the Functional Geriatric Evaluation (FGE) [12]. Before this point, there is need to identify some of the limitations of existing tools which are being used in the assessment to determine various dimensions of functioning or frailty in older populations.
Method:
Comment 1: How do you justify the data as it includes both pre-covid-19 and covid-19 period.
Comment 2: The details of measures need to be further elaborated such as which questions were presumed to measure which aspect of frailty. The details of scale used for scoring individual items.
Comment 3: Line 119: “We decided not to eliminate questions with items with a factor loading >|.30|” Please justify what is the basis for this decision and support it with guidelines from literature on statistical analysis advice for EFA.
Comment 4: Give more details of your approach for using EFA keeping in view you data was categorical with Yes and No response categories on individual items.
Results:
Comment 1: There is need to revise the headings for Table 2 and Table 3 as they are not clear. It should demonstrate the statistics that are shown in the table to report findings.
Comment 2: In Table 3, why (,) have been used instead of decimal point (.) to report factor loadings. Also the footnote should be given in end of table 3 to describe factor 1, 2 and 3.
Discussion:
The discussion is not exhaustive. Most of the points for the discussion has been touched superficially. They can be elaborated to get the complete idea of the implications for that point of discussion. The language for discussion also needs to be improved slightly e.g. line 195 “Of course, many other questionnaires assess multidimensional frailty considering socio-economic features” using the word of course is a casual way. It important to adopt scientific style of writing when discussing findings. Please also discuss some of the possible limitations of the scale because the items on the questionnaire cannot completely determine the psychological aspects of frailty e.g. only one items that is related to energy and motivation may not be accurate indicator of psychological frailty.
Conclusion: Conclusion should be differentiated from discussion. The points mentioned in this conclusion seems more related with discussion section and should be moved there by synchronizing them with relevant ideas in the discussion section.
There is a need to write a new conclusion paragraph which reflects key inferences drawn from the study and its implication with mention of some cautions that need to be considered while interpreting these findings.
Author Response
Reviewer 1
Dear Reviewer,
We appreciate your helpful comments that have allowed us to significantly improve our work. We have tried to respond to each one to clearly describe the study. We are available, if necessary, to provide further clarification of our work.
Introduction
Comment 1. As suggested we started a new paragraph after the sentence: “In order to foster and promote social participation and independent living of older adults, a better alignment between health, social care, built environments (e.g.: houses, cities, transportation systems) and resources such as Information and Communication Technologies (ICT) is needed” and changed the beginning of the next sentence.
Comment 2. Upon suggestion we improved the explanation of the multidimensional construct that is the result of a consensus conference (“This definition is the result of a large consensus conference involving experts from several countries around the world claiming for investigating frailty through the physical, psychological, and socio-economic domains”).
Comment 3. In the Introduction we modified the phrase “ The lack of integration in care processes, not allowing communication between different professions and communities in care and healing settings, causes an inadequacy in clinical and public health control in frailty management.”.
Comment 4. We added a new phrase in the Introduction: “However, the available short tools lack a strong social domain which is crucial when we are dealing with community-dwelling older adults to address care intervention”.
Methods
Comment 1. The paper assesses the predictive validity of the SFGE tool. Having used the tool also the COVID-19 period it has become stonger, due to its capacity to predict negative outcomes also during the pandemic. When we analyzed separately survival, hospitalization and institutionalization until 31.12.2019, we did not find differences in the results.
Comment 2. The details requested by the referee are in the supplementary material. In case of need, we can add it to the paper, even if this could affect the readability of the text.
Comment 3. Regarding this comment, we realized that we explained incorrectly, due to a misuse of terminology. Effectively, we did not consider factor loadings lower than |0.30|. Therefore we decided to delete the sentence so as not to create misunderstandings.
Comment 4. Some items include yes/no as an answer. In statistical analyses we considered yes/no as an ordinal variable because it adds up to the other scores, thus contributing to the final score.
Results
Comment 1. We modified the headings of Table 2 and Table 3. The statistical analyses represented in Tables 3 and 4 are described in the text:
“Through a Principal component analysis with a Varimax rotation method, we extracted 3 factors with an eigenvalue > 1.0, according to the Kaiser–Guttman rule. The Kaiser–Meyer–Olkin measure of sampling adequacy was 0.792 and with Bartlett’s test of sphericity we obtained a statistically significant result (p-value < 0.001). The initial communalities before rotation were all above 0.3, which is good. In the Rotated Factor Matrix, we did not consider factor loadings lower than |0.30| (Tables 3, 4). In conclusion, the EFA found that six items (q6, q8, q9, q10, q11, q12) had their highest loading from Factor 1 (psychophysical frailty), three items (q3, q4, q7) had their highest loading from Factor 2 (need of social and economic support) and three items (q1, q2, q5) had their highest loading from Factor 3 (lack of social relationship). Although Items q1 and q3 are both loaded on two factors (Factor 2 and Factor 3), we have associated them with Factor 3 and Factor 2, respectively, because of their greater affinity for the domain they represent. In the case of q1 and q3, we also considered the values of the items that positively loaded on the selected factor”.
Comment 2. As suggested we modified Table 3 adding zero and point. Then we added a footnote to Table 3 to explain Factors 1, 2, and 3.
Discussion
We removed the expression "of course" to improve the readability. As suggested, we added a limitation of the study at the end of the last paragraph of the discussion section. The questionnaire cannot completely determine the psychological aspects of frailty because it can be expected to provide only an initial assessment, which can be used as a basis for preventive or curative action, if necessary.
We improved the Discussion “In addition, a recent systematic review pointed out that economic difficulties play a crucial role in determining frailty in the older people. This study shows that financial problems of older adults also reducing their social network and consequently their quality of life [28]. In addition, the link between education and economic status is well known. Finally, the living arrangement is intuitively a constituent of the social dimension such that it contributes to greater or lesser frailty in this regard.
Using SFGE, frailty can be assessed on a large scale. This questionnaire can be used to screen the older population to plan public health interventions, to prevent adverse outcomes and reduce hospitalization rates.”
Conclusion
We have made the conclusion section more schematic and lightened some considerations
Reviewer 2 Report
Dear authors,
In this study, authors aimed to analise evidences of internal structure validity (not called construct validity any longer). Despite the paper presents a very relevant content, it contains several limitations that make its publication unfeasible in the way it was presented. This limitation is mostly related to psychometric procedures.
Notably, there are errors in the design of psychometric data analysis procedures. The specification of the analysis is inadequate for the type of instrument and latent variable. Principal Component Analysis (PCA) with varimax rotation has not been recommended as an Exploratory Factor Analysis (EFA) technique (despite several authors still remain using it) due to several limitations. Besides, it is not an EFA technique. Varimax rotation is not good for these type of latent variable. Robust unweighted least squares with some oblique rotation would fit better an EFA technique for the purpose of the study.
The methodological procedure of psychometric data analysis was not sufficient described and psychometric results were not sufficiently presented. Even if we were to assume the results based on a Pearson correlation matrix with PCA/varimax, the paper would still contain limitations. Several relevant informations are lacking; for example: indicators of dimensionality, results of parallel analysis, comparison between pattern matrix and rotated matrix, commonalities, quality of factor scores, .... The instrument is not well defined. I am not so sure that an instrument with 13 items sustain a 3 dimension structure. There are some cross-loadings remaining. If the adequate data extraction was used, possibly the structure would not be the same. IBM SPSS does not perform psychometric analysis based on contemporary recommendations, beginning from the matrix specification (SPSS does not have analysis for polychoric/tetrachoric matrix and does not have several relevant techniques, essencial for EFA nowadays). Factor, for example, is a software that can be used for this purpose and it is free of charges/license.
Due to all these limitations, conclusions can not be supported by results that were obtained through inadequate data analysis procedures. Because of this, I indicate the rejection of the manuscript and suggest the authors to review all the analysis specification based on contemporary psychometric procedures and recommendations. I am sorry for not being so positive at this occasion.
Best wishes,
---
Designated reviewer.
Author Response
Reviewer 2
Dear Reviewer,
We appreciate your helpful comments that have allowed us to significantly improve our work. We have tried to respond to each comment to clearly describe the analyses and their results. We are available if necessary to provide further clarification of our work.
We appreciate your comment on the methodology used for EFA. We are aware of the ongoing discussion regarding the different approaches to EFA. From a careful review of the literature (see References), it seems to us that both approaches considered, the one we proposed and the one suggested by the reviewer are reflected in the available literature. However, neither of them at the current state of the debate is to be considered as not usable in this context. Therefore, we feel that we can repropose our analytical approach while pointing out within the discussion the issues raised by the Reviewer and reported as one of the possible limitations of our work.
We are aware of the limitations of the statistical analysis because the variables are not all purely quantitative, in fact, some items have yes/no responses that we considered an ordinal variable to add to the final score. Another limitation is the cross between factor 2 and factor 3 in items q1 and q3 because we decided to use only positive associations to define the construct, in the other cases we chose those with values greater than |0.30|.
We decided to perform a Principal component analysis because the sample consisted of more than 15 people, the "missing data" was less than 1 percent, and we used pairwise deletion. Then all the needed assumptions for carrying out an EFA have been met.
We explained in the Results section: “In the correlation matrix, there were no significant correlation coefficients that suggested the presence of multicollinearity” (Supplementary Table 1). Furthermore, we added that the three identified latent Factors were psychophysical frailty (Factor 1), need for social and economic support (Factor 2), and lack of social relationship (Factor 3) with Eigenvalue of 2.94 (24.53%), 1.39 (11.58%) and 1.10 (9.21%) respectively.
References
Examples of recent papers which used “Exploratory factor analysis using principal component analysis with varimax rotation” in relation to questionnaires.
Bampa G, Kouroglou D, Metallidou P, Tsolaki M, Kougioumtzis G, Papantoniou G, Sofologi M, Moraitou D. Metacognitive Scales: Assessing Metacognitive Knowledge in Older Adults Using Everyday Life Scenarios. Diagnostics (Basel). 2022 Oct 5;12(10):2410. doi: 10.3390/diagnostics12102410. PMID: 36292099; PMCID: PMC9600082.
Sollid MIV, Slaaen M, Danielsen S, Kirkevold Ø. Psychometric properties of the person-centred coordinated care experience questionnaire (P3CEQ) in a Norwegian radiotherapy setting. Int J Qual Health Care. 2022 Sep 15;34(3):mzac067. doi: 10.1093/intqhc/mzac067. PMID: 36004618; PMCID: PMC9475430.
Nikolic A, Bukurov B, Kocic I, Soldatovic I, Mihajlovic S, Nesic D, Vukovic M, Ladjevic N, Grujicic SS. The Validity and Reliability of the Serbian Version of the Smartphone Addiction Scale-Short Version. Int J Environ Res Public Health. 2022 Jan 22;19(3):1245. doi: 10.3390/ijerph19031245. PMID: 35162268; PMCID: PMC8835088.
Wong LP, Alias H, Danaee M, Lee HY, Tan KM, Tok PSK, Muslimin M, AbuBakar S, Lin Y, Hu Z. Assessment of Impact of Containment During the COVID-19 Epidemic and Coping Behaviours Using Newly Developed Assessment Tools. Front Public Health. 2021 Dec 22;9:787672. doi: 10.3389/fpubh.2021.787672. PMID: 35004587; PMCID: PMC8728738.

Reviewer 3 Report
The review comments of “an exploratory factor analysis (EFA) of the Short Functional Geriatric Evaluation (SFGE) to assess multidimensionality of 3
frailty in community-dwelling older adults”
The purpose of this study was to develop a short evaluation scale to assess frailty in the alder adults This is a very important and interesting study. There are several problems with this study. I would like to ask the following questions.
1. The authors defined this evaluation scale as a multidimensional and short questionnaire to assess Bio-Psycho-Social frailty in older adults. However, each of the questions in the EFA is quite different from the diagnostic criteria for frailty. The questions include education, cohabitants, and finances. These questions are different from the usual frail Diagnostic Criteria.
2. The authors have already used this scale in their own studies. The authors should describe the reliability and validity that the researchers have demonstrated in previous articles, and also explain to why exploratory factor analysis is necessary in this study.
3. The factor analysis shows only factor loadings of each items. There are no other important parameters in Table1 & Table2. With regard to the results of the factor analysis, the authors should indicate precisely besides the factor loadings.
4. The percentage of participants in the survey in each area, but author should describe the reason why these areas have large range such as 0.6-49.2%.
5. SFGE is a questionnaire composed of 13 items. However, the items were reduced from 13 to 12 because questions 7 and 8 were merged in this study. Author should describe in detail the reasons for combining questions 7 and 8.
Author Response
Dear Reviewer,
We appreciate your helpful comments that have allowed us to significantly improve our work. We have tried to respond to each comment to clearly describe the study. We are available if necessary to provide further clarification of our work.
Comment 1. Thank you for the observation that allows us to expand on an important point. The items investigating education, living together, and finances are part of the social and socioeconomic domains; in fact, through the question on living together and finances we can assess social isolation and the need for socio-economic support, the question on education highlights the social network; in general, both the socioeconomic domain and the social network domain are domains proper to the multidimensional definition of Bio-Psycho-Social frailty in older people. To mention among the papers in the literature, there is a systematic review (Hayajne et al, 2021) that reports 37 papers published between 2010 and 2020 that support the link between frailty and economic status. In addition, the link between education and economic status is well known. Finally, living arrangement is intuitively a constituent of the social dimension such that it contributes to greater or lesser frailty in this regard.
Comment 2. We described the validity of the SFGE in the introduction (“This questionnaire has an acceptable short-term sensitivity (90.4%) and a medium-low specificity (78.3%) for the assessment of frailty similar to other available tools devoted to assessing frailty quickly in community-dwelling older adults”). In addition, we added in the introduction the sentence: “However, the available short tools lack a strong social domain which is crucial when we are dealing with community-dwelling older adults to address care intervention”. In fact, in this study, we have shown that the structure of the SFGE is multifactorial by identifying 3 factors: psychophysical frailty, the need for social and economic support, and the lack of social relationships.
Comment 3. To better explain "the factor loadings", at the end of the results part we added the sentence: “In the case of q1 and q3, we also considered the values of the items that were positively loaded on the selected factor”.
Comment 4. The areas have a large range because we used convenience sampling. In fact, the purpose of our study is not to assess the difference between Italian cities but to evaluate the SFGE in a general population of community-dwelling older people.
Comment 5. In 2.2. Statistical Analysis we added a paragraph to explain why the items were reduced from 13 to 12 because questions 7 and 8 were merged in this study “. The questions were both about economical difficulties, but the critical item was number 7. The item is divided into three response grades that correspond to making it to the end of the month on one's economic resources (score 0), having problems paying for a person to help at home when needed (score 1), and having problems buying necessary food (score 2). In this sense, combining the two questions into a single answer expresses the logic of the questionnaire, which wants to know the presence and extent of economic difficulties. .”

Round 2
Reviewer 2 Report
Dear authors,
Thank you for your response letter. Despite your considerations, I keep my recommendation of rejection for the reasons previously stated.
I'm sorry for not being so positive at this occasion.
Best,
Designated reviewer.
Author Response
Dear reviewer please find attached the file with our comments
with our best regards
Giuseppe Liotta

Reviewer 3 Report
The revised version has been corrected in response to peer review comments.
This manuscript is ready for publication.
Author Response
Thank you very much for supportive comments that allow us to improve the quality of the paper